# Outcomes of hyperglycaemia in pregnancy in Africa: Systematic review and meta-analysis

Ezekiel Musa[1,2,3,4]*, Tawanda Chivese[5], Mahmoud Werfalli[2,6], Larske M. Soepnel[7,8,9], Veronique Nicolaou[7], Mushi Matjila[10], Shane A. Norris[7,11], Naomi S. Levitt[1,2]*

1 Division of Endocrinology, Department of Medicine, University of Cape Town, Cape Town, South Africa, 2 Chronic Disease Initiative for Africa, Department of Medicine, University of Cape Town, Cape Town, South Africa, 3 Department of Internal Medicine, Kaduna State University, Kaduna, Nigeria, 4 North Cumbria Integrated Care NHS Foundation Trust, Carlisle, United Kingdom, 5 Department of Science and Mathematics, School of Interdisciplinary Arts and Sciences, University of Washington Tacoma, Tacoma, Washington, United States of America, 6 Department of Family and Community Medicine, Faculty of Medicine, University of Benghazi, Benghazi, Libya, 7 SAMRC/Wits Developmental Pathways for Health Research Unit, Department of Paediatrics, University of the Witwatersrand, Johannesburg, South Africa, 8 Division of Epidemiology and Biostatistics, School of Public Health, University of Cape Town, Cape Town, South Africa, 9 Julius Center for Health Sciences and Primary Care, University Medical Center Utrecht, Utrecht University, Utrecht, The Netherlands, 10 Department of Obstetrics and Gynaecology, University of Cape Town, Cape Town, South Africa, 11 School of Human Development and Health, University of Southampton, Southampton, United Kingdom

* ezemusa2000@gmail.com (EM); naomi.levitt@uct.ac.za (NSL)

## Abstract

### Objective

The global prevalence of type 2 diabetes mellitus has significantly risen in recent decades, leading to a corresponding increase in the incidence of diabetes-complicated pregnancies. Hyperglycaemia in pregnancy (HIP), the most common metabolic complication encountered during pregnancy, is associated with a range of adverse maternal and foetal outcomes. This systematic review comprehensively examined the maternal, foetal, neonatal, childhood, and long-term maternal outcomes of HIP in Africa.

### Methods

A systematic review of all studies investigating HIP outcomes in Africa from January 1998 to February 2025 was undertaken. We searched PubMed-MEDLINE, Cochrane Library, Scopus, CINAHL (EBSCOhost), Embase and Web of Science databases for eligible studies. Studies were included if they were observational studies describing outcomes of HIP in Africa. For each outcome, study results were synthesised using an inverse variance heterogeneity meta-analysis with the Freeman-Tukey transformation. Heterogeneity was assessed using the $I^2$ statistic, and publication bias was assessed using Doi plots.

**Data availability statement:** relevant data are within the paper and its Supporting Information files.

**Funding:** The author(s) received no specific funding for this work.

**Competing interests:** The authors have declared that no competing interests exist.

## Results

Thirty studies were included in the review, comprising 9742 participants. These studies were conducted across the following African countries: South Africa (n = 11), Ethiopia (n = 4), Nigeria (n = 3), Sudan (n = 3), Uganda (n = 2), and one each from Ghana, Algeria, Morocco, Democratic Republic of Congo, Zimbabwe, Togo, and Egypt. The most common adverse pregnancy outcomes for gestational diabetes mellitus (GDM) were caesarean section (CS) (overall prevalence 46.0%, 95% CI 35.7–56.4, $I^2$ = 95.6%), preterm delivery (overall prevalence 25.2% (95% CI 12.7–40.2, $I^2$ = 96.7%) and neonatal intensive care unit (NICU) admission (overall prevalence 25.9% (95% CI 13.7–40.2, $I^2$ = 85.7%). The most common adverse pregnancy outcomes for women with preexisting type 1 diabetes (T1DM) were CS (overall prevalence 57.5%, 95% CI 44.9–69.7, $I^2$ = 81.2%), preterm delivery (overall prevalence 50.7%, 95% CI 16.3–84.8, $I^2$ = 92.6%), and neonatal hypoglycaemia (overall prevalence 20.2%, 95% CI 0.0–61.4, $I^2$ = 94.6%). CS (overall prevalence 60.6%, 95% CI 45.5–74.8, $I^2$ = 93.6%) and preterm delivery (overall prevalence 35.2%, 95% CI 29.5–41.1, $I^2$ = 49.3%) were the most prevalent adverse pregnancy outcomes for women with preexisting type 2 diabetes (T2DM). Postpartum T2DM was the most common long-term adverse outcome of women who had GDM or hyperglycaemia first detected in pregnancy (HFDP). There was significant heterogeneity across most outcomes.

## Conclusions

The prevalence of adverse outcomes of HIP in Africa is high, in particular CS, preterm delivery and neonatal hypoglycaemia, with higher frequencies in pregestational T1DM and T2DM compared to GDM. Additionally, T2DM prevalence in women post-GDM is about 50%. The outcome data predominantly come from a few studies, indicating the necessity for more high-quality research to improve HIP-related maternal and child health in Africa. The high heterogeneity across most outcomes suggests that their prevalence varies across populations and underscores the need for more high-quality data. PROSPERO Registration Number: CRD42020184573.

## Introduction

While the number of people with diabetes globally has increased considerably over the past few decades, so too has the incidence of diabetes-complicated pregnancies [1]. Hyperglycaemia in pregnancy (HIP), defined by the WHO in 2014, includes diabetes first detected at any time during pregnancy, sub-classified as overt diabetes in pregnancy (DIP), gestational diabetes (GDM) and preexisting diabetes [2]. HIP is regarded as the most common metabolic complication encountered during pregnancy [2]. This is in part driven by the rising prevalence of type 2 diabetes (T2DM) and its risk factors and by changes in diagnostic criteria for GDM [3–6]. The latest global

estimates reported by the International Diabetes Federation (IDF) in 2021 indicated that 15.6% of live births to women of reproductive age (16–49 years) were complicated by HIP, with known or previously undiagnosed type 1 diabetes (T1DM) and T2DM accounting for about 16% [1,7].

Although the extent of the burden of HIP in Africa is unknown, a recent systematic review reported that the pooled prevalence of T2DM among women of childbearing age across the continent as 7.2% [8]. Further, a systematic review of the prevalence of GDM in Africa found a pooled prevalence of 13.6%, with the prevalence in Sub-Saharan Africa reported at 9% and 14.8% by different groups [9,10].

HIP is associated with adverse maternal and foetal outcomes during pregnancy. These include increased caesarean delivery rate, preeclampsia, difficult labour, macrosomia, shoulder dystocia, increased perinatal mortality, neonatal hypoglycaemia and congenital malformations [11–14]. In addition, there are accumulating global data on the long-term impact of GDM on maternal health, childhood adiposity and glucose tolerance [15–17]. There are isolated reports on pregnancy outcomes of women with pregestational diabetes in Africa [18–20]. In a systematic review of the burden, risk factors and outcomes of GDM pregnancies in SSA, Natamba et al. found that up to 2018, 6 studies reported pregnancy-related outcomes and that GDM was associated with an increased risk of macrosomia (RR: 2.19, 95% CI: 1.08–4.43) and a non-significant risk of caesarean delivery [10].

Many countries on this continent are experiencing an increasing prevalence of non-communicable diseases (NCDs), compounded by persistent high rates of infectious diseases and weak health systems. The rates of complications of different types of HIP are largely unknown, creating a significant knowledge gap essential for informed, effective health systems planning and strengthening in Africa. This systematic review examined the maternal, foetal, neonatal, childhood, and long-term maternal and offspring outcomes of HIP in Africa. Specifically, the study estimated the prevalence of adverse pregnancy outcomes for the main types of HIP in Africa, i.e., preexisting T1DM and T2DM, and GDM.

## Methods

This systematic review was conducted according to the recommendations of Cochrane Systematic Reviews, and our findings are reported in accordance with the Preferred Reporting Items for Systematic Reviews and Meta-analyses (PRISMA). The study was prospectively registered in the international database of prospectively registered systematic reviews (PROSPERO CRD42020184573), and the study protocol was published in a peer-reviewed journal [21].

### Design

This research utilised a systematic review of all literature published between January 1998 and February 2025. A meta-analysis was carried out where sufficient data with low heterogeneity were available.

### Inclusion criteria

Studies reporting the outcomes of HIP among women resident in Africa published between January 1998 and February 2025 were included as current criteria for the diagnosis of diabetes and have been widely accepted since 1998. Participants were included irrespective of age, ethnicity, educational and socioeconomic status, gestational age, and study setting. Diagnosis of pregestational diabetes (T1DM or T2DM) and GDM was defined according to WHO 1999/2013, ADA and IADPSG diagnostic criteria or definition [22–25].

All published and unpublished population-based studies, cohort or cross-sectional studies and baseline data from randomised controlled trials conducted in Africa reporting on the prevalence of HIP outcomes were included. Published multicentre studies involving African patients were also included. Studies that included either hospital settings or community settings were included. Studies were excluded if they included non-human participants, were carried out outside Africa or in Africans in the diaspora, did not contain primary data, or were qualitative studies.

## Study outcomes

**Primary Outcomes.** Primary outcomes included both maternal short-term outcomes (preeclampsia, caesarean delivery) and foetal/neonatal short-term outcomes, specifically macrosomia, congenital anomalies, intrauterine foetal death, shoulder dystocia, neonatal morbidity (hypoglycaemia, sepsis, respiratory immaturity, jaundice, neonatal ICU admission and duration of neonatal hospital stay), perinatal mortality (early neonatal death (ENNDs)) and stillbirth (SB) rates.

**Secondary outcomes.** Secondary outcomes included: 1. maternal short-term outcomes of miscarriage, preterm birth, antepartum and puerperal sepsis, gestational hypertension; 2. foetal/neonatal outcomes of birth trauma, small for gestational age (SGA) babies, and infant death; 3. long-term maternal outcomes of T2DM, metabolic syndrome, CVD risk factors; 4. long-term offspring outcomes of childhood overweight, obesity, pre-diabetes and diabetes, in addition to NCDs in adulthood (hypertension, diabetes, coronary heart disease, peripheral artery disease and cerebrovascular accidents).

**Definition of outcomes.** Definitions for key outcomes, where different criteria exist, are given below. 1. Preeclampsia: a multisystem progressive disorder characterised by the new onset of hypertension and/or worsening hypertension superimposed on chronic hypertension and proteinuria or the new onset of hypertension and significant end-organ dysfunction with or without proteinuria in the last half of pregnancy or postpartum [26,27]. 2. Primary caesarean delivery: the delivery by caesarean section (CS) for the first time [28]. 3. Congenital malformations: any single or multiple defects of the morphogenesis of organs or body regions identifiable at birth or during intrauterine life [29]. 4. Spontaneous abortion/ miscarriage; any pregnancy loss before the 28th week of gestation (this is relevant for low and middle-income countries (LMIC) settings) or loss of foetus less than 500g [30–32]. 5. Perinatal mortality: a combination of stillbirths and early neonatal death (death before 7 days) (this is relevant for an LMIC setting) [31,33].

## Search strategy for the identification of relevant studies

Using the updated African search filter [34], a sensitive search strategy was applied to retrieve all published studies of diabetes outcomes in pregnancy which were indexed in PubMed-MEDLINE, Cochrane Library, Scopus, CINAHL (EBSCO-host), Embase and Web of Science databases. There was no language restriction imposed on the literature searches. Our search strategy utilised Medical Subject Headings (MeSH) and free text. Unpublished literature was sought from experts in the field. At the same time, grey literature, such as reports, was also reviewed for relevant information from other organisational websites such as WHO, IDF, Google Scholar, and Pan African Clinical Trials Registry (PACTR). The support of an experienced Librarian was sought to validate and cross-examine our search strategy.

## Study selection for this review

Two reviewers (EM and TC) independently screened titles and abstracts of articles to identify eligible studies. Discrepancies and disagreements were addressed via group discussion and consultation between the two reviewers (EM, TC). A third reviewer (NL) arbitrated when needed. Where necessary, further information was sought from the studies' authors. Reasons for excluding articles were recorded.

## Data extraction

The Burden of Disease (BOD) Review Manager, developed by the South African Medical Research Council, was utilised to extract and record research data by four independent reviewers (EM and LS) and TC and VL [35]. The data extracted include study details:- publication date, title, design, period and objective; population- country, setting, and sample size; response rate; case definition of outcomes reported in the study; and study population characteristics. After the data extraction, the two reviewers addressed the identified differences or consulted a third reviewer (NL). Where there was missing information, the study's corresponding author was contacted to request the missing details. No response to a

maximum of three emails over two weeks sent to the corresponding author to request additional information led to the exclusion of the study. For studies appearing in more than one published article, we considered the most recent, comprehensive and one with the largest sample size. For surveys appearing in one article with multiple surveys conducted at different time points, we treated each survey as a separate study. For multinational studies, data were separated to show the country-level estimate**.**

### Assessment of risk of bias of studies

Two reviewers (MW and EM) independently evaluated and reported on the risk of bias as described in the Cochrane Handbook for Systematic Reviews of Interventions according to the criteria and associated categorisations contained therein for randomised trials and using the Risk of Bias in Nonrandomized Studies of Interventions (ROBINS-I) tool for non-randomised studies [36,37]. A consensus was reached after discussion and consultation with a third reviewer (NL).

A checklist for observational studies, adapted from the risk of bias tool for population-based studies and the Newcastle-Ottawa Scale, was used to assess the quality of non-randomised studies with standardisation in the BOD review manager.

### Data synthesis and heterogeneity assessment

Crude numerators and denominators from the individual studies were used to recalculate the study-specific prevalence. Prevalence estimates were summarised by geographic region and outcome. A meta-analysis was performed on variables that were similar across the included studies.

For the meta-analysis, variances of proportions were stabilised using the Freeman-Tukey double arcsine transformation [38,39], and then the inverse variance heterogeneity model was used to synthesise overall prevalence estimates. Heterogeneity was assessed using Cochrane's Q p-value and quantified by the $I^2$ statistic [40]. Analysis was carried out by the type of HIP, i.e., preexisting T1DM and T2DM, and GDM. The publication bias was assessed using Egger's test and Doi and funnel plots [41]. All analyses were performed using the "Metan" package in Stata version 18. Results were reported as proportions with corresponding 95% confidence intervals (CIs). Heterogeneity was assessed using the $I^2$ statistic and categorised as low heterogeneity (<50%), moderate heterogeneity (50–75%), and high heterogeneity (>75%).

### Ethics

Ethical approval was not required for this study, as it was a systematic review that utilised only published data.

## Results

### Selected Studies

The search strategy identified 3138 studies published between January 1998 and February 2025. Forty-five studies remained after removing duplicates and screening titles and abstracts. A total of 30 studies met the inclusion criteria and were included in the meta-analysis after excluding full texts with the incorrect study population, outcomes not included in the protocol, outdated diagnostic criteria for GDM, and systematic reviews (Fig 1).

### Characteristics of Included Studies

The 30 published studies included in the review had a total of 9742 participants (Table 1). These studies were conducted across the following countries (South Africa (n = 11), Ethiopia (n = 4), Nigeria (n = 3), Sudan (n = 3), Uganda (n = 2), Ghana (n = 1), Algeria (n = 1), Morocco (n = 1), Democratic Republic of Congo (n = 1), Zimbabwe (n = 1) Togo (n = 1), and Egypt (n = 1). Various study types were included in the review. The number of participants in each study type were: retrospective cohort (n = 6124), prospective cohort (n = 2868), cross-sectional (n = 440), and randomised control trials (n = 310). Overall, adverse outcomes for HIP (i.e., combined pregestational T1DM and T2DM) and GDM were reported in 5202 participants.

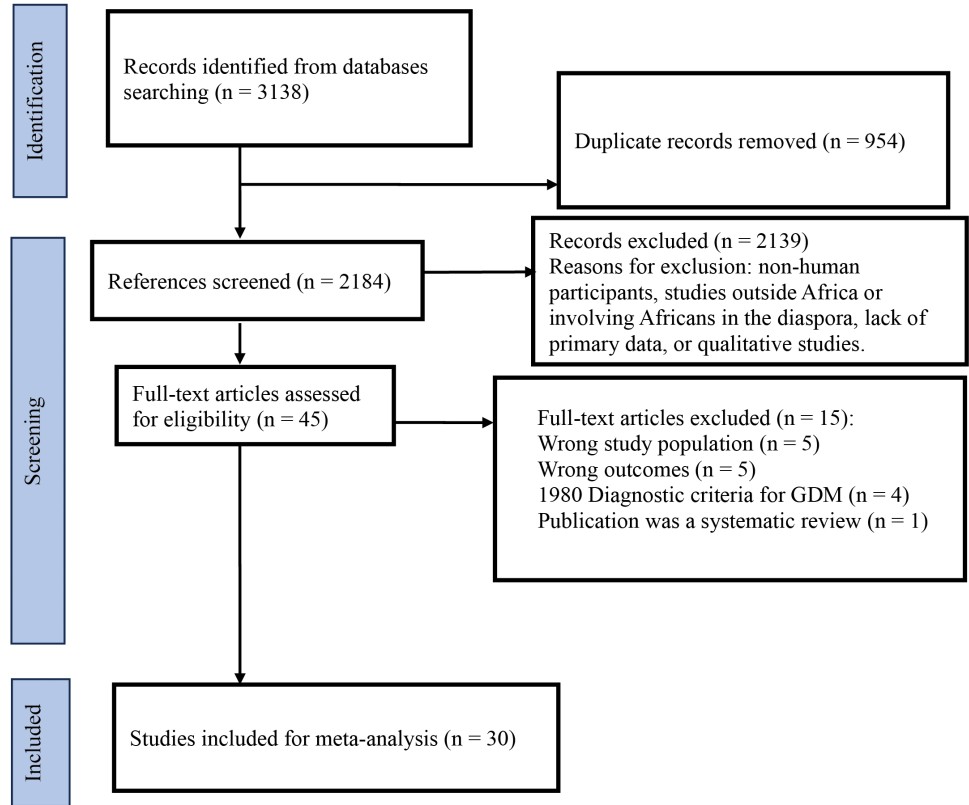

**Fig 1. Flow diagram of the included studies for the systematic review on outcomes of hyperglycaemia in pregnancy in Africa.**

Studies which reported data on GDM alone had 3544 participants. Meanwhile, only one study each reported outcomes for GDM/ hyperglycaemia first detected in pregnancy (HFDP) (n = 443) and T2DM (n = 379), respectively. These studies reported various maternal and neonatal short-term pregnancy adverse outcomes, 13 of which are summarised in this review and are among the most common in the literature.

The studies in this review used different diagnostic criteria for GDM, which included the International Association of Diabetes and Pregnancy Study Groups Consensus Panel, 2010 (IADPSG; n = 5), American Diabetes Association (ADA; n = 3), NICE 2008 (n = 3), WHO 2010 (n = 5), WHO 1998/1999 (n = 7) and other (n = 1), while diagnostic criteria were not available for 6 studies. Fifteen studies in the review were retrospective, and some included studies, which defined one or more adverse outcomes; however, definitions and/or cut-offs varied across studies.

## Quality assessment of included studies

**Risk of bias for non-RCT studies.** A summary of the risk of bias for the 28 non-RCT studies using the ROBINS-I tool is illustrated in S1 Table A. A serious risk of bias was assessed for all studies for the following domains: confounding, selection of participants, classification of interventions, measurement of outcomes, and selection of the reported results. As a result, 15 studies had severe risks of bias; for 9 studies, the risks of bias were moderate, and 4 studies had a low risk of bias. Most of the studies included in this review were retrospective and could not control for confounders. All studies were included for analysis despite their risk of bias rating.

**Table 1. Characteristics of included studies.**

| Citation (First Author, et al) | Year of publication | Country | Setting (Hospital or community-based studies) | Study design | Exposure (T1DM, T2DM, GDM, HFDP) | Sampling strategy | Total sample size | Response rate | Comparison: non-diabetic (yes/no) | Diagnostic criteria (WHO 1999/2013, ADA, IADPSG) |
|---|---|---|---|---|---|---|---|---|---|---|
| Abdelgadir et al. [42] | 2002 | Sudan | Hospital | Prospective, case control | T1DM, T2DM and GDM | Random/convenience | 88 | N/A | yes (50 controls) | WHO 1998 |
| Bawah et al. [43] | 2019 | Ghana | Hospital | Retrospective case control | GDM n = 80 | N/a | 200 | N/A | yes | ADA |
| Bhorat et al. [44] | 2019 | South Africa | Hospital | Prospective, cross-sectional study | GDM n = 54 | Consecutive for diabetes/Random for controls | 108 | | yes | WHO |
| Chivese et al. [45] | 2021 | South Africa | Hospital | Retrospective | GDM, HFDP | Total | 443 | 49.7 | no | Modified NICE 2008/WHO 2013 |
| Chivese et al. [46] | 2019 | South Africa | Hospital | Cross-sectional | GDM | Random | 220 | | no | NICE 2008/WHO 2013 |
| Chivese et al. [46] | 2019 | South Africa | Hospital | Cross-sectional | GDM | Random | 220 | | no | NICE 2008/WHO 2013 |
| Coetzee et al. [47] | 2018 | South Africa | Hospital | Prospective | GDM | Consecutive | 78 | | no | NICE 2008 |
| Feleke et al. [48] | 2020 | Ethiopia | Hospital | Retrospective and prospective cohort | GDM | Probability sampling | 1840 | | yes | RBG |
| Kheir et al. [49] | 2012 | Sudan | Hospital | Observational prospective | GDM, type 1 and 2 diabetes | Not mentioned | 50 | 100 | no | – |
| Magadla et al. [50] | 2019 | South Africa | Hospital | Retrospective medical record review | GDM, type 1 and 2 diabetes | Sub-dm population | 234 | 88.5 | no | WHO 1999 |
| Maged et al. [51] | 2016 | Egypt | Hospital | Randomised control trial – control included | GDM | Web-based randomisation | 100 | | no | ADA |
| Mimouni-Zerguini et al. [52] | 2009 | Algeria | Hospital | Prospective | GDM | Not mentioned | 150 | | yes | WHO |
| Muche et al. [53] | 2020 | Ethiopia | Hospital | Prospective cohort | GDM | Not mentioned | 131 | 92.4 | yes | ADA 2017/WHO 2013/IASDPG |
| Muche et al. [54] | 2020 | Ethiopia | Hospital | Prospective cohort | GDM | Not mentioned | 118 | | yes | ADA 2017/WHO 2013/IASDPG |
| Mukona et al. [55] | 2018 | Zimbabwe | Hospital | Prospective | GDM, type 1 and 2 diabetes | Consecutive | 157 | | no | – |
| Nakabuye et al. [56] | 2017 | Uganda | Hospital | Prospective | GDM | Not mentioned | 80 | 92 | yes | IADPSG/WHO 2013 |
| Odar et al. [57] | 2004 | Uganda | Hospital | Prospective | GDM | Consecutive | 30 | | yes | WHO 1999 |
| Opara et al. [18] | 2010 | Nigeria | Hospital | Prospective | GDM, type 1 and 2 diabetes | Consecutive | 47 | | yes | – |
| Ozumba et al. [58] | 2004 | Nigeria | Hospital | Retrospective medical record review | GDM, type 1 and 2 diabetes | Sub-dm population | 207 | 96.6 | yes | WHO 1999 |
| Soepnel et al. [59] | 2019 | South Africa | Hospital | Retrospective medical record review | GDM, type 1 and 2 diabetes | Not mentioned | 1071 | | no | IADPSG/WHO 2013 |

*(Continued)*

**Table 1.** (Continued)

| Citation (First Author, et al) | Year of publication | Country | Setting (Hospital or community-based studies) | Study design | Exposure (T1DM, T2DM, GDM, HFDP) | Sampling strategy | Total sample size | Response rate | Comparison: non-diabetic (yes/no) | Diagnostic criteria (WHO 1999/2013, ADA, IADPSG) |
|---|---|---|---|---|---|---|---|---|---|---|
| Tandu-Umba et al. [60] | 2012 | Democratic Republic of Congo | Hospital | Prospective | GDM | Not mentioned | 108 | | no | |
| Utz et al. [61] | 2018 | Morocco | Hospital/Community health centres | Randomised control trial (screening and detection) | GDM | Not mentioned | 210 | | no | IADPSG/WHO 2013 |
| Van Zyl et al. [20] | 2018 | South Africa | Hospital | Retrospective descriptive | GDM, type 1 and 2 diabetes | Not mentioned | 725 | | no | WHO 1999 |
| Dafallah et al. [62] | 2004 | Sudan | Hospital | Prospective case-control | GDM (n = 230) and preexisting diabetes (n = 130), IGT n = 330 | Random | 1280 | | yes | WHO |
| Daponte et al. [63] | 1999 | South Africa | Hospital | Retrospective audit | T1DM, T2DM, GDM (pregestational defined as insulin-requiring and non-insulin requiring | Random | 142 | | no | 100gm 3 hour |
| Djagadou et al. [64] | 2019 | Togo | Hospital | Retrospective audit | GDM | Audit | 125 | | no | 75gm OGTT 2hr |
| Ekpebegh et al. [19] | 2006 | South Africa | Hospital | Retrospective | T2DM only | Audit | 379 | | no | N/A |
| Bajrond et al. [65] | 2019 | Ethiopia | | Retrospective, cross-sectional | All | Audit | 346 | | no | N/A |
| Huddle et al. [66] | 2005 | South Africa | Hospital | Retrospective analysis | T1DM, T2DM, GDM (pregestational defined as insulin-requiring and non-insulin-requiring | Audit | 733 | | no | 75gm OGTT WHO |
| John et al. [67] | 2015 | Nigeria | Hospital | Retrospective analysis | All | Audit | 122 | | yes | 75gm OGTT WHO |
| Nicolaou et al. [68] | 2022 | South Africa | Hospital | Prospective cohort study | HFDP | Not mentioned | 103 | | yes | 75gm OGTT IADPSG |

ADA: American Diabetes Association, GDM: Gestational diabetes mellitus, HFDP: Hyperglycaemia first detected in pregnancy, IADPSG: International Association of the Diabetes and Pregnancy Study Groups, IGT: Impaired glucose tolerance, NICE: National Institute for Health and Care Excellence, N/A: Not available, OGTT: Oral glucose tolerance tests, T1DM: Type 1 diabetes, T2DM: Type 2 diabetes, WHO: World Health Organisation

**Risk of bias for RCT studies.** A summary of the risk of bias of included RCT-selected studies using the Cochrane tool is shown in S1 Table B. One study (Maged et al. 2016) failed to report random sequence generation and allocation concealment details [51]. Only one study (Utz et al. 2018) described blinding of participants. Further, the outcomes assessor was reported to be blinded in the same study (Utz et al. 2018) [61]. Only one study (Utz et al. 2018) described

the reasons for participants' withdrawals [61]. In the GRADE analysis, three studies were judged as having high quality (Maged et al. 2016; Utz et al. 2018; Muche et al. 2020) [51,53,61]. Ten studies were considered moderate quality, and 15 were considered low or deficient quality (S2 Table).

## Outcomes in pregnancies with GDM

As seen in Table 2, the most common reported pregnancy outcomes for GDM were CS (overall prevalence 46.0%, 95% CI 35.7–56.4, $I^2 = 95.6\%$), preterm delivery (overall prevalence 25.2%, 95% CI 12.7–40.2, $I^2 = 96.7\%$) and NICU admission (with overall prevalence 25.9%, 95% CI 13.7–40.2, $I^2 = 85.7\%$). The least prevalent outcomes were miscarriage, neonatal death, congenital malformation, stillbirth, perinatal death and shoulder dystocia. Other adverse outcomes reported included macrosomia (overall prevalence 17.7%, 95% CI 12.4–23.7, $I^2 = 92.2\%$), neonatal hypoglycaemia(overall prevalence 15.0%, 95%CI 6.8–25.5, $I^2 = 95.0\%$), neonatal jaundice, (overall prevalence 12.4%, 95%CI 0.0–36.7, $I^2 = 90.4\%$), preeclampsia (overall prevalence 11.7%, 95%CI 4.1–22.1, $I^2 = 94.5\%$), pregnancy-induced hypertension (overall prevalence 11.3%, 95%CI 4.7–20.0, $I^2 = 91.0\%$), and respiratory distress syndrome (RDS with overall prevalence of 7.3%, 95%CI 5.0–10.0, $I^2 = 0.0\%$) (Table 2). There was significant heterogeneity across all the outcomes (Table 2).

Subgroup analysis by country showed that the prevalence of CS was highest in South Africa (62.5%, 95%CI 53.6–71.0, $I^2 = 90.3\%$), with lower but varied prevalence across the other countries (S1 File). The prevalence of preterm birth was also highest in South African studies (33.2%, 95%CI 24.1–43.1, $I^2 = 85.3\%$). There was no notable variation across the other countries for the remaining outcomes (S1 File).

**Table 2. Overall synthesis of maternal-foetal outcomes for GDM.**

| Outcomes | Range of raw prevalence estimate (%) | Overall prevalence % (95% CI) | No. of studies | No. of participants (n) | I² (%) |
|---|---|---|---|---|---|
| **Maternal outcomes** | | | | | |
| CS | 20.0 - 82.8 | 46.0 (35.7-56.4) | 14 | 4524 | 95.6 |
| preeclampsia | 2.4 - 22.4 | 11.7 (4.1-22.1) | 5 | 2418 | 94.5 |
| PIH | 1.9 - 23.8 | 11.3 (4.7-20.0) | 5 | 1904 | 91.0 |
| Miscarriage | 0.0 - 1.6 | 1.4 (0.8-2.2) | 4 | 2218 | 0.0 |
| **Offspring outcomes** | | | | | |
| Macrosomia | 8.0–62.1 | 17.7 (12.4-23.7) | 13 | 4906 | 92.2 |
| Neonatal hypoglycaemia | 0.0–62.1 | 15.0 (6.8-25.5) | 12 | 2578 | 95.0 |
| Preterm delivery | 1.0–84.6 | 25.2 (12.7-40.2) | 9 | 2632 | 96.7 |
| Neonatal RDS | 3.7–15.8 | 7.3 (5.0-10.0) | 5 | 820 | 0.0 |
| Congenital malformation | 0.0–11.1 | 2.0 (0.9-3.4) | 9 | 3076 | 52.0 |
| Stillbirth | 0.0–16.7 | 2.9 (1.7-4.4) | 9 | 3984 | 62.9 |
| Neonatal death | 0.0–8.0 | 1.5 (0.3-3.6) | 8 | 2888 | 78.5 |
| Neonatal jaundice | 3.0–48.3 | 12.4 (0.0-36.7) | 4 | 350 | 90.4 |
| Perinatal death | 0.0–14.5 | 3.9 (1.1-8.0) | 7 | 3402 | 91.5 |
| Shoulder dystocia | 0.8–23.3 | 3.8 (0.0-11.8) | 3 | 1472 | 90.3 |
| NICU admission | 11.0–54.5 | 25.9 (13.7-40.2) | 4 | 688 | 85.7 |

CS: Caesarean section, PIH: Pregnancy-induced hypertension, RDS: Respiratory distress syndrome, NICU: Neonatal intensive care unit. For the meta-analysis, proportions were stabilised using the Freeman-Tukey double arcsine transformation, and then the inverse variance heterogeneity model was used to synthesise overall prevalence estimates.

## Outcomes in pregnancies with T2DM

Caesarean section (overall prevalence 60.6%, 95% CI 45.5–74.8, $I^2$=93.6%), preterm delivery (overall prevalence 35.2%, 95% CI 29.5–41.1, $I^2$=49.3%), and neonatal RDS (prevalence 19.4%, 95% CI 7.1–35.4, one study) were the most prevalent adverse pregnancy outcomes for women with preexisting T2DM. The least prevalent adverse outcomes were shoulder dystocia, neonatal death, congenital malformation, miscarriage and stillbirth, all with prevalence below 3% (Table 3). Again, there was significant heterogeneity across all outcomes, except for the related outcomes of perinatal death, miscarriage, stillbirth and outcomes with single studies. However, there were not enough studies to explore heterogeneity by subgroup.

## Outcomes in pregnancies with T1DM

The most common adverse pregnancy outcomes for women with preexisting T1DM were CS (overall prevalence 57.5%, 95% CI 44.9–69.7, $I^2$=81.2%), preterm delivery (overall prevalence 50.7%, 95% CI 16.3–84.8, $I^2$=92.6%), neonatal hypoglycaemia (overall prevalence 20.2%, 95%CI 0.0–61.4, $I^2$=94.6%), neonatal jaundice (overall prevalence 16.1%, 95% CI 4.9–31.5, one study) and neonatal RDS (prevalence 13.2%, 95% CI 4.0–26.1, one study). Similar to outcomes for T2DM, the least prevalent outcomes were neonatal death, congenital malformation, and shoulder dystocia (Table 4). There was considerable heterogeneity across all outcomes, except for the related outcomes of stillbirth, perinatal death and neonatal death. There was also low heterogeneity for two other related outcomes, macrosomia and shoulder dystocia. As with T2DM, there were not enough studies to explore the sources of heterogeneity in the outcomes, given the high $I^2$ values.

## Long-term maternal and foetal outcomes for HIP

Only a few studies reported data on medium- to long-term outcomes after pregnancies complicated by HFDP, so no meta-analyses were conducted for these outcomes. The most frequently reported long-term maternal outcome among

**Table 3. Overall synthesis of maternal-foetal outcomes for T2DM.**

| Outcomes | Range of raw prevalence estimate (%) | Overall prevalence (95% CI) | No. of studies | No. of participants (n) | $I^2$ (%) |
|---|---|---|---|---|---|
| **Maternal outcomes** | | | | | |
| CS | 25.8–78.3 | 60.6 (45.5-74.8) | 4 | 1580 | 93.6 |
| Preeclampsia | 3.2–19.4 | 8.0 (1.9-17.0) | 3 | 1266 | 87.5 |
| PIH | 7.6 | 7.6 (5.2-10.4) | 1 | 816 | 0.0 |
| Miscarriage | 2.6–2.9 | 2.9 (1.7-4.4) | 2 | 1204 | 0.0 |
| **Offspring outcomes** | | | | | |
| Macrosomia | 8.2–8.5 | 8.5 (6.6-10.6) | 3 | 1480 | 0.0 |
| Neonatal hypoglycaemia | 7.5–22.6 | 11.9 (0.7-30.6) | 2 | 490 | 81.9 |
| Preterm delivery | 32.7–38.7 | 35.2 (29.5-41.1) | 2 | 1104 | 49.3 |
| Neonatal RDS | 19.4 | 19.4 (7.1-35.4) | 1 | 62 | 0.0 |
| Congenital malformation | 0.7–6.5 | 2.2 (0.4-5.1) | 4 | 1440 | 73.1 |
| Stillbirth | 1.9–3.2 | 2.9 (1.8-4.2) | 3 | 1508 | 0.0 |
| Neonatal death | 0.9–12.5 | 2.0 (0.2-5.1) | 4 | 894 | 52.6 |
| Neonatal jaundice | 10.5 | 10.5 (2.4-22.6) | 1 | 76 | 0.0 |
| Perinatal death | 4.1–6.1 | 4.9 (3.4-6.5) | 3 | 1508 | 0.0 |
| Shoulder dystocia | 0.0 | 0.0 (0.0-0.9) | 1 | 388 | 0.0 |

CS: Caesarean section, PIH: Pregnancy-induced hypertension, RDS: Respiratory distress syndrome, NICU: Neonatal intensive care unit. For the meta-analysis, proportions were stabilised using the Freeman-Tukey double arcsine transformation, and then the inverse variance heterogeneity model was used to synthesise overall prevalence estimates.

**Table 4. Overall synthesis of maternal-foetal outcomes for T1DM.**

| Outcomes | Range of raw prevalence estimate (%) | Overall prevalence (95% CI) | No. of studies | No. of participants (n) | I² (%) |
|---|---|---|---|---|---|
| **Maternal outcomes** | | | | | |
| CS | 31.6–67.1 | 57.5 (44.9-69.7) | 4 | 800 | 81.2 |
| preeclampsia | 2.5–21.1 | 10.7 (0.0-31.7) | 3 | 550 | 90.7 |
| PIH | 11.4 | 11.4 (7.3-16.2) | 1 | 404 | 0.0 |
| Miscarriage | 5.7–6.9 | 6.9 (4.0-10.5) | 2 | 474 | 0.0 |
| **Offspring outcomes** | | | | | |
| Macrosomia | 5.2–8.6 | 7.0 (4.6-10.0) | 3 | 696 | 0.0 |
| Neonatal hypoglycaemia | 9.9-42.1 | 20.2 (0.0-61.4) | 2 | 420 | 94.6 |
| Preterm delivery | 34.5–68.6 | 50.7 (16.3-84.8) | 2 | 400 | 92.6 |
| Neonatal RDS | 13.2 | 13.2 (4.0-26.1) | 1 | 76 | 0.0 |
| Congenital malformation | 0.6–5.7 | 2.0 (0.5-4.3) | 4 | 720 | 30.2 |
| Stillbirth | 4.1–5.7 | 5.0 (2.9-7.5) | 3 | 712 | 0.0 |
| Neonatal death | 0.0–2.6 | 0.4 (0.0-2.7) | 3 | 490 | 35.9 |
| Neonatal jaundice | 16.1 | 16.1 (4.9-31.5) | 1 | 62 | 0.0 |
| Perinatal death | 5.7–7.0 | 7.0 (4.5-9.9) | 3 | 712 | 0.0 |
| Shoulder dystocia | 2.9–6.7 | 4.6 (0.2-12.5) | 2 | 100 | 0.0 |

CS: Caesarean section, PIH: Pregnancy-induced hypertension, RDS: Respiratory distress syndrome, NICU: Neonatal intensive care unit. For the meta-analysis, proportions were stabilised using the Freeman-Tukey double arcsine transformation, and then the inverse variance heterogeneity model was used to synthesise overall prevalence estimates.

women with GDM or HFDP was progression to T2DM. Four studies reported the prevalence of T2DM after HFDP, with three from South Africa reporting a prevalence of 21%−48%, and a single study from Ethiopia reporting a prevalence of 6.8%. One South African study reported a metabolic syndrome prevalence of 40.8% in women with prior HFDP. One South African study assessed long-term outcomes in offspring and reported a prevalence of 26.5% for overweight/obesity at preschool age (Table 5).

## Discussion

This systematic review and meta-analysis, consisting of 30 studies from 12 countries in Africa, revealed that HIP is associated with high rates of adverse maternal and foetal outcomes. The most predominant maternal pregnancy outcome of HIP was CS, while the commonest adverse foetal outcomes were preterm delivery, neonatal hypoglycaemia, and neonatal

**Table 5. Long-term maternal foetal outcomes for HIP.**

| Outcomes | Prevalence (%) | No. of participants (n) | HIP type | Country | Authors (year) |
|---|---|---|---|---|---|
| **Maternal outcome** | | | | | |
| T2DM | 6.8 | 10355 | GDM | Ethiopia | Feleke et al. (2020) [48] |
| | 48 | 220 | HFDP | South Africa | Chivese et al. (2019) [69] |
| | 21 | 78 | GDM | South Africa | Coetzee et al. (2018) [47] |
| | 44.6 | 103 | HFDP | South Africa | Nicolaou et al. (2022) [68] |
| Metabolic Syndrome | 40.8 | 103 | HFDP | South Africa | Nicolaou et al. (2022) [68] |
| **Offspring outcomes** | | | | | |
| Overweight/obesity | 26.5 | 167 | HFDP | South Africa | Chivese et al. (2021) [45] |

HFDP: Hyperglycaemia first detected in pregnancy.

jaundice. Few studies have reported long-term outcomes for the mother and/or offspring with considerable heterogeneity in the overall prevalence across most of the outcomes.

While CS was the most common adverse outcome in HIP, the overall prevalence was lower among women with GDM (46.0%) than among women with T2DM (60.6%) or T1DM (57.5%). These findings are consistent with those from studies conducted outside of Africa. However, there are conflicting reports about the prevalence of CS in GDM compared to pregestational diabetes in Africa. For example, Huddle et al. reported higher rates of CS among women with GDM compared to pregestational T1DM but similar rates in pregestational T2DM [70–74]. The high rates of CS in pregnant women with diabetes in Africa are similar to global figures. This signals the widespread adoption of global surgical strategies to mitigate the human and infrastructure challenges the continent faces in providing safe and timely CS. Generally, clinical practice is to induce labour in women with HIP but no macrosomia at 38 weeks (on treatment) or 40 weeks (on dietary control) and to proceed to CS where there is no active labour or failure of progression of labour. Women with HIP and macrosomia are delivered via elective CS to obviate the risk of birth complications such as obstructive labour and shoulder dystocia. The high rates of CS partly reflect this clinical practice, limited obstetric resources, and early elective CS for diabetes. Additionally, effective management of major contributing factors to CS, such as maternal weight gain, obesity and preeclampsia, is essential.

The prevalence of preterm delivery was highest in T1DM, followed by T2DM, and lastly, GDM, following the same trend as for CS. These findings are consistent with the systematic review conducted by Malaza et al., who found six studies reporting higher rates of preterm birth in pregestational T1DM and T2DM than in GDM [75]. However, Stogianni et al. reported the highest prevalence of preterm delivery in T2DM, followed by T1DM and then GDM [71]. Whether the severity of hyperglycaemia and poor glycaemic control are potential factors underlying the higher frequency of premature delivery in pregestational diabetes types compared to GDM is speculative, and the actual mechanism underpinning these findings requires further investigation. Preterm delivery, defined as the delivery of a baby before 37 completed weeks of pregnancy, is associated with higher morbidity and mortality in infants, particularly in Africa, where most facilities and expertise for managing premature babies are inadequate [76]. Furthermore, the significant prevalence of premature delivery associated with HIP in Africa poses a greater risk of long-term poor neurodevelopment, behavioural difficulty and cognitive disability in surviving preterm children [77,78]. This highlights the need to improve the management of HIP in Africa by implementing specialised services encompassing pregnancy and early neonatal life.

Neonatal hypoglycaemia, defined as plasma glucose of less than 2.5 mmol/L, is a significant cause of morbidity in newborns and a preventable cause of poor neurodevelopment [79]. This was most common in T1DM in this review, in keeping with a study by Yamamoto et al [73]. The placental transfer of exogenous insulin from the mother to the foetus is the likely major mechanism. In HIP, in general, hyperinsulinaemia resulting from the placental transport of nutrients, mainly glucose, from the mother to the foetus, is proposed to be the primary physiological mechanism underlying neonatal hypoglycaemia in offspring [80,81]. Interestingly, neonatal hypoglycaemia was more common in GDM than in T2DM. The 2023 American Diabetes Association (ADA) guidelines utilising findings from a real-world study of a Swedish T1DM cohort and the CONCEPTT trial in women with TIDM suggest that real-time continuous glucose monitoring (rt-CGM) can decrease the incidence of neonatal hypoglycaemia in pregnancy complicated by GDM [82–84]. Similarly, the National Institute for Health and Care Excellence (NICE) guidelines in 2020 and 2023 Korean Diabetes Association suggest that rt-CGM should be offered to all pregnant people with type 1 diabetes [85,86]. In keeping with this suggestion, Yu et al. reported a significantly lower frequency of neonatal hypoglycaemia in GDM women who used CGM compared to standard blood glucose monitoring [87]. These data point to the inclusion of CGM in the glycaemic management of pregnant women with HIP, regardless of the type of diabetes, to prevent and or easily identify and correct maternal hypoglycaemia, thereby limiting unnecessary overuse of insulin and resultant hypoglycaemia in neonates, however in most LMIC including in Africa the limited access to and costs of CGM will see very few women receive this technology.

Similar to findings in neonatal hypoglycaemia, neonatal jaundice was most common in T1DM, followed by GDM and lastly, T2DM. Neonatal hyperbilirubinaemia, associated with a shortened erythrocyte lifespan, could be explained by

hyperglycaemia-associated lipid peroxidation, which induces changes in red cell membrane fluidity and structural alterations [88]. Additional significant contributors to neonatal jaundice in infants of mothers with diabetes are poor glycaemic control and decreased polyunsaturated fatty acids (PUFAs) [89], while potential intrauterine hypoxia and neonatal sepsis are other potential factors [90]. Neonatal jaundice is associated with bilirubin-induced neurologic dysfunction (BIND) due to neurotoxicity to critical areas in the brain, with other complications of neonatal jaundice, including failure to thrive and deficiency of fat-soluble vitamins [91].

Intriguingly, NICU admissions were very common in GDM, with approximately 26% of 688 infants in four studies from South Africa, Egypt, Algeria and Uganda being admitted to the NICU. The reasons for admission, however, were not provided. Notably, no studies reported on NICU admissions in offspring of women with T1DM and T2DM [44,51,52,56]. Studies conducted outside the African continent have reported high rates of NICU admissions in T1DM (55%) and T2DM (66.7%), with lower admission rates in GDM [73,92]. Our findings of common NICU admissions in infants of women with GDM call for synergistic and collaborative care of women with GDM as well as their newborns to minimise mortality, particularly in Africa, where facilities for NICU care are limited. It was surprising that there were no reports of NICU admission in T1DM and T2DM, suggesting an interesting area to interrogate in future research. Other expected short-term adverse outcomes of HIP in this systematic review were preeclampsia, PIH, macrosomia, and neonatal RDS. The prevalence of macrosomia in women with GDM in our review was 17.7%, which appears to be lower than in high-income countries, where macrosomia among women with GDM is commonly reported between 20–30%. These differences may be largely due to high rates of maternal obesity, excessive gestational weight gain, or widespread use of IADPSG diagnostic criteria in higher-income countries rather than higher biological risk [22,93]. HIP was associated with miscarriage, stillbirth, perinatal death, neonatal death, congenital malformation and shoulder dystocia, with a prevalence of less than 5%.

In this systematic review, only seven studies reported long-term maternal-foetal outcomes of HIP. Three studies reported a wide range in the prevalence of T2DM in HIP of between 6.8% and 48%, from 6 weeks up to 10 years postpartum. The highest prevalence was reported by Chivese et al., particularly in women with HFDP. These data support the wide range in the incidence of T2DM among women with GDM, as reported in the systematic review by Kim et al., which ranged from 2.6% to over 70% across 28 studies from 6 weeks to 28 years postpartum [94]. A number of studies/systematic reviews reported a peak increase in T2DM in the first 5 years postpartum, with a plateau at 10 years after delivery [94]. Recently, another systematic review by Vounzoulaki et al. reported that T2DM was approximately 10 times higher in women with past GDM than in healthy controls and T2DM cumulative incidences of 16.46%, 15.58% and 9.91% in women of mixed ethnicity, non-white, and white populations, respectively [95]. Similarly, Juan et al. reported that GDM women diagnosed with IADPSG criteria were about 6 times more likely to progress to T2DM than the controls, with a cumulative incidence of T2DM of 12.1% [96]. The risk of T2DM in women with GDM is most significant within the first five years after pregnancy and plateaus at about ten years, with fasting plasma glucose levels >5.9 mmol/L during pregnancy being the most significant predictor of future T2DM development [94,97]. Given that the risk of developing T2DM is highest in the first 5 years postpartum, it is important to ensure that diabetes is assessed at regular intervals: within the first 6 weeks and then annually, to enable early diagnosis and management. Our study further provides data suggesting that T2DM may potentially be regarded as an epidemic, particularly in Africa, where health systems are weak, non-communicable diseases are poorly managed, and there are high personal and societal costs of treatment. This potential public health disaster requires a multidisciplinary approach to mitigating the rising prevalence of T2DM. Current trials suggest that postpartum, T2DM progression can be reduced with both lifestyle and pharmacological interventions [98]. Consequently, there is a profound need for well-designed controlled trials utilising either pharmacological or lifestyle modification interventions for T2DM prevention in GDM or non-GDM women in Africa.

Our findings of the high prevalence of overweight/obesity among offspring of women with HFDP are in keeping with the increased risk of metabolic syndrome in studies [69]. In a systematic review and meta-analysis conducted by Pathirana et al., GDM-exposed offspring in utero had a greater risk of developing metabolic syndrome in comparison with non-GDM exposed

offspring in utero [99]. Similarly, Wan Mahmud Sabri et al. found that approximately 70% of the children with metabolic syndrome were born to mothers with GDM and that children born to non-GDM women had about a 50% reduction in the odds of having metabolic syndrome in comparison to those born to GDM mothers [100]. The mechanisms underpinning overweight/obesity and metabolic syndrome include foetal hyperinsulinaemia, metabolic memory and epigenetic changes. GDM is characterised by hyperinsulinemia-induced foetal overgrowth and gluco- and lipotoxic milieu, creating a cascade of metabolic programmes, which may persist postpartum, leading to obesity and or metabolic syndrome in the offspring [101–103]. Furthermore, Sauder et al. showed that children exposed to intrauterine hyperglycaemia had increased Homeostatic Model Assessment of Insulin Resistance (HOMA-IR) irrespective of their body mass index [104]. Meanwhile, Kelstrup et al. demonstrated a lower expression of the peroxisome proliferator-activated receptor-γ coactivator-1α (PPARGC1A), a major mediator of insulin sensitivity, in skeletal muscle of offspring exposed to GDM [105]. These childhood cardiovascular risk factors can be significantly prevented by effective care of women with HIP and lifestyle modifications in their offspring.

This review has a number of limitations. Some outcome findings were based on only a few studies, with many confounding factors; outcomes were not stratified by treatment due to inconsistent reporting. Even when meta-analysis was performed, considerable heterogeneity remained due to clinical and methodological diversity across the included studies; therefore, interpretation should be cautious. This is likely because different studies could have reported different prevalences of pregnancy adverse outcomes due to varying distributions of maternal obesity, hypertension, age or parity. This could have been better addressed using subgroup analyses rather than adjusted estimates or meta-regression. However, most of the studies we included did not provide subgroup estimates for any of these variables; therefore, we could not address this concern. Furthermore, most included studies originated from South Africa, where health system capacity, screening practices, and treatment protocols differ from those in other African countries; therefore, the generalisability of the results is limited. This underscores the necessity for more robust, high-quality research on the outcomes of HIP in Africa. In particular, future research should evaluate effective interventions to prevent progression to T2DM among African women with GDM, examine how social determinants such as rural residence and limited antenatal care shape HIP outcomes, and assess the cost-effectiveness of universal versus risk-based screening in resource-constrained settings."

In conclusion, the prevalence of adverse outcomes of HIP in Africa is high. Notably, T2DM prevalence in women who have had GDM is up to 50% by 6 years, highlighting the urgent need for well-designed prospective cohort studies with systematic postpartum screening to better characterise long-term risks. Substantial heterogeneity was observed in estimates across most outcomes, suggesting variation in the prevalence of the adverse outcome across settings and populations. Concerted efforts by healthcare professionals, governments and charitable organisations are required to improve care for women with HIP to curtail the high rates of their complications. Effective pre-conception counselling and care for T1DM and T2DM are essential; however, these are often absent or poorly implemented in the region. Additionally, optimising metabolic control during pregnancy, combined with multidisciplinary care of all women with pregestational diabetes and GDM, is crucial in reducing adverse outcomes of HIP. Importantly, screening for diabetes in women post-GDM, initially from 6 weeks to 6 months and then annually for the first 5 years, is essential for early diagnosis and management. Similarly, postpartum interventions for mothers to decrease the risk of developing T2DM and lifestyle changes to prevent cardiometabolic risk in the offspring are crucial.

## Research in context

### Why was this study done?

- The pooled prevalence of GDM in Africa was reported as 13.6%.

- Limited reports exist on pregnancy outcomes for women with pregestational diabetes in Africa, with GDM linked to increased macrosomia and a non-significant rise in caesarean deliveries in a systematic review.

- What is the prevalence of adverse outcomes of hyperglycaemia in pregnancy (HIP) in Africa?

## What did the researchers do and find?

We conducted a systematic review and meta-analysis of outcomes of HIP in Africa, including studies published from January 1998 to February 2025.

- The study provides updated evidence on the prevalence of maternal, foetal, neonatal, childhood and long-term maternal and offspring outcomes of HIP in Africa.
- We found a high prevalence of adverse outcomes of HIP, including increased rates of caesarean sections, preterm deliveries, neonatal hypoglycaemia, and jaundice.
- For the first time, notably, up to 50% of women with previous GDM develop type 2 diabetes within six years. Metabolic syndrome is the most common long-term outcome among offspring of women who had hyperglycaemia during pregnancy.

## What do these findings mean?

◦ The findings emphasise the importance of more coordinated efforts among healthcare professionals, policymakers, and organisations to improve care for women with HIP and reduce complications, ultimately enhancing maternal and child health in Africa. Additionally, there is a pressing need for more high-quality research to further advance HIP-related maternal and child health in the region.

## Strengths and limitations of this study

- The study provided updated evidence on the prevalence of maternal, foetal, neonatal, childhood and long-term maternal and offspring outcomes of hyperglycaemia in pregnancy in Africa.
- The study adhered to the Preferred Reporting Items for Systematic Reviews and Meta-analyses (PRISMA).
- Meta-analysis was not possible for some specific outcomes due to a limited number of included studies.
- The review exhibited significant heterogeneity across most outcomes, indicating considerable variability in study methods, populations, and effect estimates, which limits the generalisability of the findings.

## Supporting information

**S1 Table. A and B. The Risk of Bias in Non-Randomised Studies – (ROBINS-I) assessment tool and Risk of bias for included RCTs.**
(DOCX)

**S2 Table. Assessment of quality of evidence (GRADE) in the included studies.**
(DOCX)

**S1 File. Supplementary subgroup analyses.**
(PDF)

**S2 File. HIP SR data extraction_combined.**
(XLSX)

**S3 File. HIP SR results-metaanalysis.**
(PDF)

**S4 File. PRISMA_2020_checklist.**
(DOCX)

## Author contributions

**Conceptualization:** Ezekiel Musa, Naomi S Levitt.

**Data curation:** Ezekiel Musa, Tawanda Chivese, Mahmoud Werfalli, Larske M Soepnel, Veronique Nicolaou.

**Formal analysis:** Tawanda Chivese, Mahmoud Werfalli.

**Methodology:** Ezekiel Musa, Tawanda Chivese, Mahmoud Werfalli, Larske M Soepnel, Veronique Nicolaou, Mushi Matjila, Naomi S Levitt.

**Software:** Ezekiel Musa.

**Supervision:** Mushi Matjila, Shane A Norris, Naomi S Levitt.

**Writing – original draft:** Ezekiel Musa.

**Writing – review & editing:** Ezekiel Musa, Tawanda Chivese, Mahmoud Werfalli, Larske M Soepnel, Veronique Nicolaou, Mushi Matjila, Shane A Norris, Naomi S Levitt.

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
