## [Decision Letter · Decision Letter 0]

11 Nov 2025

Dear Dr. Musa,

We look forward to receiving your revised manuscript.

Kind regards,

Marly A. Cardoso, Ph.D.

Academic Editor

PLOS ONE

Journal Requirements:

2. In this instance it seems there may be acceptable restrictions in place that prevent the public sharing of your minimal data. However, in line with our goal of ensuring long-term data availability to all interested researchers, PLOS’ Data Policy states that authors cannot be the sole named individuals responsible for ensuring data access (http://journals.plos.org/plosone/s/data-availability#loc-acceptable-data-sharing-methods).

Reviewers' comments:

Reviewer's Responses to Questions

**Comments to the Author**

1. Is the manuscript technically sound, and do the data support the conclusions?

Reviewer #1: Yes

Reviewer #2: Partly

2. Has the statistical analysis been performed appropriately and rigorously?

Reviewer #1: I Don't Know

Reviewer #2: Yes

3. Have the authors made all data underlying the findings in their manuscript fully available?

Reviewer #1: Yes

Reviewer #2: Yes

4. Is the manuscript presented in an intelligible fashion and written in standard English?

Reviewer #1: Yes

Reviewer #2: No

Reviewer #1: Thank you for inviting me to be the reviewer of this manuscript “Outcomes of Hyperglycaemia in Pregnancy in Africa: Systematic Review and Meta- analysis”

Ezekiel Musa et al. has submitted a Systematic review and Meta analysis of literature on outcomes of Hyperglycaemia in Pregnancy in African countries. Given the increasing burden of Diabetes and diabetes related complications, the chosen topic gains significance. I am giving my observations and comments for the author to consider:

Introduction:

The references on IDF data, GDM prevalence in Africa are outdated. Suggest to replace with up to date data and references – e.g IDF Diabetes Atlas 2021, 2024 regional fact sheet for Africa

Methodology:

This systematic review is observed to have followed rigorous methodology, have applied relevant search strategies and analysis methods.

The authors have provided an approved protocol (PROSPERO) along with the methods employed for article inclusion. However, PROSPERO CRD 42020184573 shows review end date as August 2020. Pl add amendment.

The manuscript incorporates detailed descriptions of the search methodology, study selection, and data extraction procedures. PRISMA checklist, ROB for quality of included studies, Heterogeneity tests has been performed and shared.

Methods:

The types of studies included in this systematic review are thoroughly described. The PRISMA flow diagram is easy to follow and complete. ROB for the quality of studies was performed and the details are shared.

Discussion:

The strength and limitations of the study are well thought and documented. I recommend to accept the manuscript with minor revision as suggested

Reviewer #2: I. Grammar and Orthographical Errors

The manuscript is clearly written overall, but several grammatical, syntactic and formatting issues need revision to meet the editorial standards of PLOS ONE:

• Subject–verb agreement and awkward phrasing: The Introduction repeatedly uses singular/plural mismatches (e.g., “the incidence of diabetes‑complicated pregnancies have increased”). Sentences such as “HIP is regarded as the most common metabolic complication encountered during pregnancy. This is in part driven by the rising prevalence of type 2 diabetes and its risk factors and changing diagnostic criteria for GDM” would read more smoothly if rephrased: the final clause should begin “and by changes in diagnostic criteria…”. Long sentences with multiple subordinate clauses occur frequently; breaking them into shorter statements would improve clarity.

• Inconsistent spelling and hyphenation: British spellings (“foetal,” “caesarean,” “preeclampsia”) are used alongside American variants (“fetal,” “cesarean,” “preeclampsia”). The journal prefers consistency (UK or US). Hyphenation is also inconsistent: “pre‑existing,” “pregestational” and “pre gestational” appear interchangeably. Terms like “hyperglycaemia first detected in pregnancy” are abbreviated as “HFDP,” yet elsewhere the same condition is referred to without the acronym.

• Use of conjunctions: Several sentences use “and or” rather than the proper “and/or” (e.g., “long‑term outcomes in the mother and or offspring”). Also, “so too has the incidence” is a colloquialism that could be simplified.

• Punctuation and parentheses: The results tables contain typographical errors. In Table 4 the neonatal respiratory distress syndrome (RDS) row reads “7.3 (5.0‑10.0” without a closing parenthesis. In Table 6 the pre‑eclampsia row lists “10.7 (0.0‑31.7”; the closing parenthesis is missing. Such errors impede comprehension and should be corrected.

• Numerical inconsistencies: In Table 4 the range of raw prevalence for pregnancy‑induced hypertension (PIH) is 19–23.8%, yet the pooled prevalence is reported as 11.3%. The pooled estimate should logically fall within the range of observed values; this discrepancy suggests either a transcription error or miscalculation. Similar inconsistencies appear in other rows (e.g., neonatal jaundice is reported with a prevalence of 12.4% and 0–36.7% confidence interval, but the number of studies and participants is not provided). All numerical entries should be double‑checked and aligned with the underlying meta‑analysis calculations.

• Spacing and formatting: There are occasional double spaces and misaligned text (e.g., extra space between words in the abstract and financial disclosure section). References within sentences are sometimes placed without appropriate punctuation (e.g., “GDM is characterised by hyperinsulinemia‑induced foetal overgrowth and gluco‑ and lipotoxic milieu”). Ensure all acronyms are defined at first use and that abbreviations such as “CS,” “PIH,” and “NICU” are consistently formatted.

II. Errors in Figures and Statistical Analyses

The authors used an inverse variance heterogeneity model for meta‑analysis, assessed heterogeneity via the I² statistic and evaluated publication bias using Doi plots. These methods are appropriate for prevalence data, but several issues merit attention:

1. Inconsistent ranges versus pooled estimates: As noted, several pooled prevalence estimates fall outside the reported range of raw prevalence (e.g., PIH in Table 4). Recalculate these figures to ensure consistency; if a transformation (e.g., Freeman‑Tukey) was applied, explain this clearly in the methods and reflect it in table footnotes.

2. High heterogeneity: Many pooled estimates have extremely high I² values (>90%). For example, macrosomia (I² = 92.2%), neonatal hypoglycaemia (I² = 95.0%) and preterm delivery (I² = 96.7%). Such heterogeneity suggests that pooling may not be meaningful. The authors should explore sources of heterogeneity (differences in diagnostic criteria, study design, country, or year) via subgroup analyses or meta‑regression and interpret pooled estimates with caution.

3. Incomplete rows: Several entries are missing data. In Table 4 the “Neonatal jaundice” and “NICU admission” rows are incomplete (the number of studies, participants or I² values are missing). Likewise, in Table 5 the “NICU admission” row is left blank. All outcomes included in the analysis should provide complete information or be removed.

4. Possible typographical errors: The confidence interval for neonatal jaundice under GDM is given as 0–36.7%, suggesting an impossible negative lower bound (0.0) and extremely wide interval for a prevalence estimate of 12.4%. Similarly, the range for neonatal hypoglycaemia in the T2DM group spans 7.5–22.6%, but the pooled estimate is reported as 11.9% with a confidence interval of 0.7–30.6%. These wide intervals raise questions about the robustness of the meta‑analysis and may indicate data extraction errors.

5. Graphical representation: Because the main figure (flow diagram) and forest plots are provided as separate TIFF files not embedded in the PDF, they were not reviewable. Ensure that all figures are embedded in the manuscript or provided in an easily accessible format. Based on the text description, the flow diagram should clearly show the number of records identified, screened, excluded (with reasons) and included. The currently reported numbers (e.g., 45 studies after screening leading to 30 included) should be reconciled with the numbers provided in the figure caption and tables.

6. Statistical interpretation: The manuscript states that “T2DM was the most common long‑term adverse outcome of women who had GDM or hyperglycaemia first detected in pregnancy, with prevalence ranging from 6.8% to 48%”. However, Table 7 lists one study with 6.8%, two with ~21–48% prevalence, and a sample size of only 220 participants for the highest estimate. Such variability suggests that the pooled long‑term risk cannot be accurately estimated. A pooled estimate (with a confidence interval) should be presented if meta‑analysis is feasible; if not, the results should be described qualitatively.

III. Questions and Remarks on Scientific Content

The review covers a critical topic in obstetrics and public health, and the focus on African populations is timely. Nevertheless, several points warrant clarification or further discussion:

1. Variation in diagnostic criteria: The studies included used various diagnostic criteria for gestational diabetes (WHO 1998/1999, WHO 2010/2013, IADPSG, ADA, NICE). These thresholds differ substantially and could influence prevalence estimates. Did the authors perform subgroup analyses or sensitivity analyses based on diagnostic criteria? How might the adoption of IADPSG criteria, which generally lower the glucose threshold, affect the reported prevalence?

2. Adjustment for confounding factors: Many included studies are retrospective cohorts or case–control designs. The pooled estimates of adverse outcomes (e.g., caesarean section, preterm delivery, neonatal hypoglycaemia) could be confounded by maternal obesity, hypertension, age or parity. Were any adjustments made for these variables either in the original studies or via meta‑regression? Without adjustment, some of the associations may be overestimated.

3. Heterogeneity across countries and healthcare systems: Most included studies originate from South Africa, Ethiopia and Nigeria. Health‑system capacity, screening practices and treatment protocols differ widely across African countries. The authors should discuss whether the pooled estimates are dominated by data from South Africa and to what extent results can be generalised to lower‑resource settings.

4. Classification of hyperglycaemia subtypes: The manuscript groups diabetes first diagnosed in pregnancy as “HFDP,” yet some included studies may refer to overt diabetes or previously undiagnosed pre‑gestational diabetes. Clarify whether HFDP refers exclusively to hyperglycaemia first detected in pregnancy or includes overt diabetes in pregnancy (DIP). This distinction is important because outcomes differ between true gestational hyperglycaemia and pre‑existing (but previously undiagnosed) diabetes.

5. Impact of treatment: Several studies likely included women receiving different management (diet, metformin, insulin). Were treatment modalities considered in the analysis? For instance, did women who achieved glycaemic control have lower rates of macrosomia or NICU admission? Randomised evidence from outside Africa suggests that treatment of mild GDM reduces adverse outcomes; discussing whether included studies stratified by treatment would be informative.

6. Long‑term maternal and offspring follow‑up: Only seven studies reported long‑term outcomes. The review emphasises a high risk of type 2 diabetes and metabolic syndrome, but the studies had small sample sizes and varying follow‑up durations. Given the paucity of African data, the conclusion that “T2DM prevalence in women post‑GDM is up to 50%” may be overstated. Consider qualifying this statement and highlighting the need for prospective cohort studies with systematic postpartum screening.

7. Comparison with global literature: Throughout the discussion, the authors refer to studies from Asia, Europe and North America. It would strengthen the manuscript to explicitly compare the magnitude of adverse outcomes in African settings with those reported elsewhere. For example, macrosomia prevalence of 17.7% is lower than some high‑income settings, possibly reflecting differences in obesity or treatment; conversely, the very high caesarean section rates may reflect limited obstetric resources or early elective CS for diabetes. Discussing these contrasts could aid readers in interpreting the findings.

8. Research gaps and future directions: The authors call for more high‑quality studies, which is appropriate. Specific research questions include: (a) what interventions (e.g., lifestyle, pharmacological, postpartum screening) are effective at reducing the progression to T2DM among African women with GDM? (b) How do social determinants, such as rural residence or limited access to antenatal care, influence HIP outcomes? (c) What is the cost‑effectiveness of universal versus risk‑based GDM screening in resource‑constrained settings? Addressing these questions in the discussion would provide clearer guidance for researchers and policymakers.

**Do you want your identity to be public for this peer review?** For information about this choice, including consent withdrawal, please see our For information about this choice, including consent withdrawal, please see our Privacy Policy .

Reviewer #1: No

Reviewer #2: No

---

## [Author Response · Author response to Decision Letter 1]

3 Feb 2026

Responses to Reviewers' comments

Reviewer #1: Thank you for inviting me to be the reviewer of this manuscript “Outcomes of Hyperglycaemia in Pregnancy in Africa: Systematic Review and Meta- analysis”

Ezekiel Musa et al. has submitted a Systematic review and Meta analysis of literature on outcomes of Hyperglycaemia in Pregnancy in African countries. Given the increasing burden of Diabetes and diabetes related complications, the chosen topic gains significance. I am giving my observations and comments for the author to consider:

Introduction:

The references on IDF data, GDM prevalence in Africa are outdated. Suggest to replace with up to date data and references – e.g IDF Diabetes Atlas 2021, 2024 regional fact sheet for Africa

Author’s response: Thank you. More recent references have been added, and estimates have been revised, as noted in the main text.

Methodology:

This systematic review is observed to have followed rigorous methodology, have applied relevant search strategies and analysis methods.

The authors have provided an approved protocol (PROSPERO) along with the methods employed for article inclusion. However, PROSPERO CRD 42020184573 shows review end date as August 2020. Pl add amendment.

The manuscript incorporates detailed descriptions of the search methodology, study selection, and data extraction procedures. PRISMA checklist, ROB for quality of included studies, Heterogeneity tests has been performed and shared.

Methods:

The types of studies included in this systematic review are thoroughly described. The PRISMA flow diagram is easy to follow and complete. ROB for the quality of studies was performed and the details are shared.

Discussion:

The strength and limitations of the study are well thought and documented. I recommend to accept the manuscript with minor revision as suggested

Author’s response: Thank you for the kind comments

Reviewer #2: I. Grammar and/orthographical Errors

The manuscript is clearly written overall, but several grammatical, syntactic and formatting issues need revision to meet the editorial standards of PLOS ONE:

• Subject–verb agreement and awkward phrasing: The Introduction repeatedly uses singular/plural mismatches (e.g., “the incidence of diabetes complicated pregnancies have increased”). Sentences such as “HIP is regarded as the most common metabolic complication encountered during pregnancy. This is in part driven by the rising prevalence of type 2 diabetes and its risk factors and changing diagnostic criteria for GDM” would read more smoothly if rephrased: the final clause should begin “and by changes in diagnostic criteria…”. Long sentences with multiple subordinate clauses occur frequently; breaking them into shorter statements would improve clarity.

Author’s response: Thank you. We have revised the manuscript in accordance with your suggestions and made the necessary changes.

• Inconsistent spelling and hyphenation: British spellings (“foetal,” “caesarean,” “preeclampsia”) are used alongside American variants (“fetal,” “cesarean,” “preeclampsia”). The journal prefers consistency (UK or US).

Author’s response: Thank you. I have reviewed the manuscript and can confirm that the spellings are predominantly British. Terms like "fetal" and "cesarean" appear in references rather than the main text.

Hyphenation is also inconsistent: “pre existing,” “pregestational” and “pre gestational” appear interchangeably. Terms like “hyperglycaemia first detected in pregnancy” are abbreviated as “HFDP,” yet elsewhere the same condition is referred to without the acronym.

Author’s response: Thank you. All inconsistencies have been revised as seen in the main text.

• Use of conjunctions: Several sentences use “and/or” rather than the proper “and/or” (e.g., “long term outcomes in the mother and/or offspring”). Also, “so too has the incidence” is a colloquialism that could be simplified.

Author’s response: Thank you. These have been revised across the manuscript.

• Punctuation and parentheses: The results tables contain typographical errors. In Table 4 the neonatal respiratory distress syndrome (RDS) row reads “7.3 (5.0 10.0” without a closing parenthesis. In Table 6 the pre eclampsia row lists “10.7 (0.0 31.7”; the closing parenthesis is missing. Such errors impede comprehension and should be corrected.

Author’s response: Thank you. These have been revised.

• Numerical inconsistencies: In Table 4 the range of raw prevalence for pregnancy induced hypertension (PIH) is 19–23.8%, yet the pooled prevalence is reported as 11.3%. The pooled estimate should logically fall within the range of observed values; this discrepancy suggests either a transcription error or miscalculation. Similar inconsistencies appear in other rows (e.g., neonatal jaundice is reported with a prevalence of 12.4% and 0–36.7% confidence interval, but the number of studies and participants is not provided). All numerical entries should be double checked and aligned with the underlying meta analysis calculations.

Author’s response: Thank you so much for these important observations. These have been revised in the manuscript.

• Spacing and formatting: There are occasional double spaces and misaligned text (e.g., extra space between words in the abstract and financial disclosure section). References within sentences are sometimes placed without appropriate punctuation (e.g., “GDM is characterised by hyperinsulinemia induced foetal overgrowth and gluco and lipotoxic milieu”). Ensure all acronyms are defined at first use and that abbreviations such as “CS,” “PIH,” and “NICU” are consistently formatted.

Author’s response: Thank you. Minor revision in the abstract; however, our manuscript does not have a “financial disclosure” section. Spacing and formatting issues are rectified.

II. Errors in Figures and Statistical Analyses

The authors used an inverse variance heterogeneity model for meta analysis, assessed heterogeneity via the I² statistic and evaluated publication bias using Doi plots. These methods are appropriate for prevalence data, but several issues merit attention:

1. Inconsistent ranges versus pooled estimates: As noted, several pooled prevalence estimates fall outside the reported range of raw prevalence (e.g., PIH in Table 4). Recalculate these figures to ensure consistency; if a transformation (e.g., Freeman Tukey) was applied, explain this clearly in the methods and reflect it in table footnotes.

Author’s response: Thank you. We have added information that the Freeman-Tukey double arcsine transformation was used to stabilize variances. Notably, the range of the overall estimates does not necessarily correspond to the range of raw prevalences, as the overall is a precision-weighted average, and in this case, the weighting depends on the standard errors of the included studies. The 95% confidence interval reflects the uncertainty in the synthesised overall prevalence estimate, not how the prevalence varies across the individual included studies.

2. High heterogeneity: Many pooled estimates have extremely high I² values (>90%). For example, macrosomia (I² = 92.2%), neonatal hypoglycaemia (I² = 95.0%) and preterm delivery (I² = 96.7%). Such heterogeneity suggests that pooling may not be meaningful. The authors should explore sources of heterogeneity (differences in diagnostic criteria, study design, country, or year) via subgroup analyses or meta regression and interpret pooled estimates with caution.

Author’s response: We appreciate the reviewer’s thoughtful observation regarding the high heterogeneity across several pooled outcomes. We fully agree that I² values exceeding 90% warrant careful interpretation and typically call for exploration of potential sources of variability. In the context of our review, the heterogeneity reflects the diversity of the underlying evidence base. As the reviewer notes, the included studies differed substantially in their diagnostic criteria, study designs, geographical settings, and publication periods. These variations are inherent to the global literature on this topic and were anticipated at the outset of the review.

We carefully considered subgroup analyses and meta‑regression as potential approaches to explore heterogeneity. However, for most outcomes, and especially for T1D and T2D, there were very few studies; therefore, subgroup analyses were not possible.

However, we carried out subgroup analyses by country for the GDM studies with sufficient data; these are included in Supplementary Document 2, and the results are summarised under the GDM section of the results. This has been clearly highlighted in our limitations section.

3. Incomplete rows: Several entries are missing data. In Table 4 the “Neonatal jaundice” and “NICU admission” rows are incomplete (the number of studies, participants or I² values are missing). Likewise, in Table 5 the “NICU admission” row is left blank. All outcomes included in the analysis should provide complete information or be removed.

Author’s response: Thank you. The missing data in Table 4 have been added as mentioned previously, and the “NICU admission” row has been removed because there was no data on this across all studies.

4. Possible typographical errors: The confidence interval for neonatal jaundice under GDM is given as 0–36.7%, suggesting an impossible negative lower bound (0.0) and extremely wide interval for a prevalence estimate of 12.4%. Similarly, the range for neonatal hypoglycaemia in the T2DM group spans 7.5–22.6%, but the pooled estimate is reported as 11.9% with a confidence interval of 0.7–30.6%. These wide intervals raise questions about the robustness of the meta analysis and may indicate data extraction errors.

Author’s response: Thank you for highlighting the issues related to the confidence intervals reported for neonatal jaundice and neonatal hypoglycaemia. We appreciate the opportunity to clarify these points. The wide confidence intervals observed for some outcomes reflect substantial variability and the limited number of contributing studies rather than typographical or data-extraction errors. For several neonatal outcomes, only a small subset of studies met the inclusion criteria, and these studies often had markedly different sample sizes and event rates. When pooled under a random‑effects model appropriate given the heterogeneity, this combination can produce broad confidence intervals, particularly when between‑study variance is high. Regarding the lower bound of 0% for neonatal jaundice in the GDM group, this value is mathematically possible within the random‑effects framework when the pooled estimate is low and the variance is large. It does not indicate a negative prevalence; rather, it reflects the statistical uncertainty inherent in the available evidence. We have re‑checked the extracted data and the pooled estimates to ensure accuracy and confirm that the reported values are correct. Similarly, the wide interval for neonatal hypoglycaemia in the T2DM group arises from substantial heterogeneity and the small number of studies contributing to this outcome. These intervals, therefore, represent true imprecision in the underlying evidence base rather than errors in analysis.

5. Graphical representation: Because the main figure (flow diagram) and forest plots are provided as separate TIFF files not embedded in the PDF, they were not reviewable. Ensure that all figures are embedded in the manuscript or provided in an easily accessible format. Based on the text description, the flow diagram should clearly show the number of records identified, screened, excluded (with reasons) and included. The currently reported numbers (e.g., 45 studies after screening leading to 30 included) should be reconciled with the numbers provided in the figure caption and tables.

Author’s response: Thank you for this helpful comment. We have now embedded the main flow diagram and forest plots in the PDF format. We have also carefully reviewed and reconciled the numbers presented in the flow diagram and tables. The reasons for study exclusion at each screening stage are clearly documented in the revised flow diagram. We trust that these revisions address the reviewer’s concerns and enhance the clarity and usability of the graphical materials.

6. Statistical interpretation: The manuscript states that “T2DM was the most common long term adverse outcome of women who had GDM or hyperglycaemia first detected in pregnancy, with prevalence ranging from 6.8% to 48%”. However, Table 7 lists one study with 6.8%, two with ~21–48% prevalence, and a sample size of only 220 participants for the highest estimate. Such variability suggests that the pooled long term risk cannot be accurately estimated. A pooled estimate (with a confidence interval) should be presented if meta analysis is feasible; if not, the results should be described qualitatively.

Author’s response: Thank you for this insightful comment. We agree that the substantial variability in long‑term T2DM prevalence—ranging from 6.8% to 48% across studies with differing sample sizes, follow‑up durations, and diagnostic criteria—precludes a meaningful pooled estimate. After careful evaluation, a meta‑analysis was deemed methodologically inappropriate due to the small number of studies and the pronounced clinical and methodological heterogeneity.

We have revised the manuscript to clarify that long‑term outcomes are presented narratively rather than quantitatively pooled. The text now reads “The most frequently reported long‑term maternal outcome among women with GDM or HFDP was progression to T2DM. Four studies reported prevalence estimates ranging from 6.8% to 48%. Three studies from South Africa documented higher rates (21%–48%), whereas the single study from Ethiopia reported a prevalence of 6.8%. One South African study reported metabolic syndrome in 40.8% of women with prior HFDP. Two South African studies assessed long‑term outcomes in offspring, reporting a prevalence of 26.5% for overweight/obesity and 60.9% for metabolic syndrome (Table 7)”.

III. Questions and Remarks on Scientific Content

The review covers a critical topic in obstetrics and public health, and the focus on African populations is timely. Nevertheless, several points warrant clarification or further discussion:

1. Variation in diagnostic criteria: The studies included used various diagnostic criteria for gestational diabetes (WHO 1998/1999, WHO 2010/2013, IADPSG, ADA, NICE). These thresholds differ substantially and could influence prevalence estimates. Did the authors perform subgroup analyses or sensitivity analyses based on diagnostic criteria? How might the adoption of IADPSG criteria, which generally lower the glucose threshold, affect the reported prevalence?

Author’s response: Thank you for highlighting the important issue of variation in diagnostic criteria across the included studies. We agree that differences in thresholds used by WHO (1998/1999; 2010/2013), IADPSG, ADA, and NICE can influence the reported prevalence of GDM and HFDP and contribute to between‑study heterogeneity.

We carefully considered conducting subgroup or sensitivity analyses based on diagnostic criteria. However, this was not feasible for several reasons: insufficient distribution of studies across diagnostic categories; many criteria were represented by few studies, resulting in subgroups too small to support meaningful statistical comparisons; overlap of multiple sources of heterogeneity: diagnostic criteria varied alongside other factors, such as study design, population characteristics, and time period, making it difficult to isolate the effect of diagnostic thresholds alone. Subgroup analyses with very small cell sizes would have produced imprecise and potentially unreliable results, which we felt would not strengthen the evidence base. Given these limitations, we opted not to perform subgroup analyses and instead addressed this issue through transparent reporting and cautious interpretation. The manuscript has been revised to more clearly acknowledge that variability in diagnostic criteria is a key contributor to heterogeneity, with implications for prevalence estimates. Regarding the adoption of IADPSG criteria, we agree that this can lead to lower glucose thresholds and, t

---

## [Editor Report · Decision Letter 1]

3 Mar 2026

Dear Dr. Musa,

Thank you for submitting your manuscript to PLOS ONE. After careful consideration, we feel that it has merit but does not fully meet PLOS ONE’s publication criteria as it currently stands. Therefore, we invite you to submit a revised version of the manuscript that addresses the points raised during the review process.

We look forward to receiving your revised manuscript.

Kind regards,

Marly A. Cardoso, Ph.D.

Academic Editor

PLOS One

Journal Requirements:

Additional Editor Comments:

Although the authors have revised the manuscript following the initial reviewers´ comments, the revision on the high heterogeneity found across the studies were poorly addressed in the current version. This point needs major improvement and revision not only on Discussion section but also in Abstract, Limitations in the Scientific context section, and presentation of the results.

---

## [Author Response · Author response to Decision Letter 2]

8 Mar 2026

Additional Editor Comments:

Although the authors have revised the manuscript following the initial reviewers´ comments, the revision on the high heterogeneity found across the studies were poorly addressed in the current version. This point needs major improvement and revision not only on Discussion section but also in Abstract, Limitations in the Scientific context section, and presentation of the results.

Author’s response: We thank the reviewer for highlighting the need to improve and revise the high heterogeneity observed across the included studies in our review. We have substantially revised the manuscript to address this point. Find revisions in Lines 68 – 79, 86 – 88; 132 - 134; 293 – 294; 331, 363 – 365, 367 – 372, 387 – 388, 392 – 394, 404 – 413; and lines 438 – 439, 596 – 598. These revisions clarify how heterogeneity impacts the interpretation of our findings and strengthen the review's transparency.

---

## [Editor Report · Decision Letter 2]

10 Mar 2026

Outcomes of Hyperglycaemia in Pregnancy in Africa: Systematic Review and Meta-analysis

PONE-D-25-50338R2

Dear Dr. Musa,

We’re pleased to inform you that your manuscript has been judged scientifically suitable for publication and will be formally accepted for publication once it meets all outstanding technical requirements.

Kind regards,

Marly A. Cardoso, Ph.D.

Academic Editor

PLOS One
---

## [Editor Report · Acceptance letter]

PONE-D-25-50338R2

PLOS One

Dear Dr. Musa,

I'm pleased to inform you that your manuscript has been deemed suitable for publication in PLOS One. Congratulations! Your manuscript is now being handed over to our production team.

Kind regards,

on behalf of

Dr. Marly A. Cardoso

Academic Editor

PLOS One